# Review of Polarization Optical Devices Based on Graphene Materials

**DOI:** 10.3390/ijms21051608

**Published:** 2020-02-26

**Authors:** Shijie Zhang, Zongwen Li, Fei Xing

**Affiliations:** School of Physics and Optoelectronic Engineering, Shandong University of Technology, Zibo 255049, China; zhangsj.sdut@foxmail.com (S.Z.); zwli@stumail.sdut.edu.cn (Z.L.)

**Keywords:** graphene, polarization, optics, biological, optical fiber

## Abstract

Graphene has received extensive scholarly attention for its extraordinary optical, electrical, and physicochemical properties, as well as its compatibility with silicon-based semiconductor processes. As a unique two-dimensional atomic crystal material, graphene has excellent mechanical properties, ultra-high carrier mobility, ultra-wide optical response spectrum, and strong polarization dependence effect, which make it have great potential in new optical and polarization devices. A series of new optical devices that are based on graphene have been developed, showing excellent performance and broad application prospects. In this paper, the recent research progress of polarizers, sensors, modulators, and detectors that are based on the polarization characteristics of graphene is reviewed. In particular, the polarization dependence effect and broadband absorption enhancement of graphene under total reflection structure are emphasized, which enhance the interaction between graphene and light and then provide a new direction for research of graphene polarization devices.

## 1. Introduction

Graphene, as a new type of carbon nanomaterial, is a monoatomic layer of SP_2_ carbon atom with two-dimensional hexagonal crystal structure (HCC) [1]. Increasing attention has been paid to the research of graphene since the announcement of the 2010 Nobel Prize in Physics [2,3,4], which has become the hotspot and frontier of multi-disciplinary research. In the process of further research on the preparation method and properties of graphene, its application field is expanding. As a zero-bandgap semi-metal material, graphene has a unique Dirac tapered band structure and linear dispersion relationship, excellent force, thermal, optical, and electrical properties, extremely high carrier mobility, and unique transmission transport properties [2,4,5]. The excellent electronic transmission properties of graphene [6,7,8] make graphene have some tunable broadband optical properties [9,10]. It also makes the light less dependent on the defects and preparation process of graphene, which makes the application of graphene in optics easier.

The optical properties of graphene [4,11,12,13,14] are mainly influenced by its chemical potential. Its conductivity includes in-band conductivity and inter-band conductivity: σ=$\sigma {\;}_{inter}$ + $\sigma_{inter}^{\prime }$+$i\sigma_{inter}^{\prime \prime }$. Chemical doping or applied voltage can control the chemical potential of graphene to change its conductivity, thereby controlling the optical properties of graphene. When graphene is slightly doped, inter-band conductivity plays a leading role, and graphene is more sensitive to light in higher frequency bands. When the chemical potential of graphene satisfies |μ|>ℏω∕2 (where ℏ and ω are reduced Planck constants and incident angle frequencies, respectively), they can be excited by electromagnetic waves in the Terahertz (THz) band to generate surface plasmons [15]. Moreover, graphene oxide (GO) exhibits a fluorescent effect. Light waves that are in the range of 350 to 600 nm can effectively excite GO to generate fluorescence, and its emission wavelength increases with the increase of the excitation wavelength. Studies have shown that the fluorescence of GO is caused by electron-hole recombination from the conduction band to the valence band. From the perspective of atomic structure, it is attributable to the transfer of electrons between the non-oxidized state region and the oxidized state region. Graphene has a significant absorption of visible to infrared wavelengths (2.3%) despite being only one atomic layer thick, being 50 times the thickness of gallium arsenide. Theories and experiment both show that this is a constant value determined by the fine structure constant, independent of the frequency of incident light [16]. Although graphene exhibits higher light absorption as a single atomic layer material, as a material, the light absorption of 2.3% is still exceedingly low. This indicates that the interaction between light and graphene is still weak, and many optical properties of graphene are limited by this weaker light absorption. Therefore, a prominent problem of graphene optics is how to find the interaction between light and graphene. Here, a sandwich graphene structure is proposed, which can effectively enhance the interaction between graphene and light. Figure 1 shows the structure. This structure not only has strong optical absorption of transverse electrical (TE) waves under total reflection conditions, but it also increases the broadband absorption of graphene. In addition, graphene also has saturation absorption effect, that is, as the intensity of incident light increases, the transmittance of incident light to graphene gradually increases until saturation.

Light waves are transverse waves, and polarization is one of their important properties. It can also have different polarization states when transmitting in the same direction. In optics and optoelectronics, it is often necessary to detect the polarization characteristics of light, change the polarization state, and use the polarization characteristics to indirectly measure the changes of some physical quantities [17]. Optical polarization devices have become one of the indispensable devices in optical systems with the development of laser technology, optical fiber technology, and optical polarization technology. According to the principle of polarized light generation, traditional polarizing devices can be roughly divided into the following three categories: absorptive polarization controller using material anisotropic absorption, prism polarization controller while using refraction effect, and Brewster angle polarization controller using reflection. In 2011, Vakil and Engheta studied the electron transport characteristics on the surface of single-atom graphene [18], showing the great potential of graphene in applications, such as metamaterials and transformation optics. The imaginary part of$\;\sigma_{i}$ graphene conductivity can be obtained by adjusting the chemical potential in different frequency ranges to be negative and positive values. Setting uneven substrate materials and applying grid electric or magnetic fields, according to this property, can obtain the conductivity of graphene with uneven pattern distribution. When $\sigma_{i\;}$> 0, the graphene layer exhibits the metallic properties of surface plasmon polariton (SPP) surface waves capable of effectively supporting the transverse magnetic (TM) mode. When$\;\sigma_{i}$ < 0, graphene no longer supports SPP surface waves in TM mode. Instead, weakly guided TE mode SPP waves may occur. A series of graphene-based polarization devices have been emerging and have important roles in communication [19,20,21], biomedicine [22,23,24], imaging [25], and soon, based on the above research. In this paper, based on the polarization characteristics of graphene, the research progress of polarizing devices is selectively discussed, including polarizers, sensors, modulators, and detectors. Table 1 shows a brief overview of graphene-based polarizing devices.

## 2. Graphene-Based Polarizer

In conventional polarizers that are based on the semiconductor quantum well structure, since metals are used, they are generally designed while using Drude models. Meanwhile, the imaginary part of the conductivity of the metal is greater than 0, which can only be designed as a TM polarizer [26]. There are some in-depth researches on how to calculate the conductivity of graphene [27,28,29,30,31]. Among them, most scholars in the industry recognize the results that were published by G. Hanon [30] in 2008. He equated graphene to a two-dimensional material and calculated its surface conductivity using the Kubo formula. It was deduced that when the temperature, incident wave length, and scattering rate were determined, the conductivity of graphene was only related to the chemical potential. Graphene can obtain the properties of the medium or metal under the influence of different applied voltage because of its adjustable conductivity. Therefore, the polarization controller in TE or TM mode can be realized by using graphene, which cannot be achieved by previous materials. In this regard, the addition of graphene as a new material to the design of polarizers has become increasingly attractive.

### 2.1. Graphene Optical Fiber Polarizer

Optical fiber polarizer is a part of the most important optical passive devices in the optical fiber system, which is characterized by polarization or phase. It is widely used because of its lightweight, small footprint, high extinction ratio, and strong compatibility with optical fiber systems. At present, in the fiber optic gyroscope system and the fiber optic sensing system, it has become the key passive device of polarized light generation in the fiber optic system and it plays an irreplaceable role. Fiber optic polarizers can be divided into fiber optic attenuation polarizers and fiber optic cut-off polarizers. The polarization principle of the former is to enhance the attenuation difference of the two polarization modes. Through the propagation of a certain length, the mode with high propagation loss is eliminated by the loss, thus leaving the other mode with low propagation loss. The polarization principle of the latter is that one polarization mode end and the other pass through. Typical fiber optic polarizers include metal clad fiber optic polarizers [32,33], crystal clad fiber optic polarizers [34], and toroidal fiber optic polarizers [35]. At the same time, due to graphene’s excellent electrical and optical properties, its chemical potential can be controlled by external voltage or chemical doping. For chemical doping, the n-doping state of graphene can be obtained by metal atom doping, and the p-doping state of graphene can be achieved by polymer molecules that are composed of N, O, F, and other elements. Moreover, the substitutionally doping graphene is more stable than the surface transfer doping graphene. It should be noted that the surface coverage of the chemical species or the number of foreign atoms incorporated into the basal plane of graphene is difficult to control and the doping of graphene would be difficult to reproduce. We need exact methods to homogenously and reproducibly dope graphene p-type/n-type for different applications and control the doping level [36]. Therefore, controlling the chemical potential of graphene can effectively control the transmission of the TE mode and the TM mode.

Achieving efficient coupling and highly constrained steering is a challenge due to the atomic thickness of graphene. Bao et al. [26] proposed directly covering the structure of the fiber with graphene to achieve polarization control. Figure 2a shows the exact structure. They cut the fiber along the direction of propagation to achieve a D-type fiber structure. Graphene is then coated onto the surface of the d-type fiber, so that the graphene and the core of the fiber can contact each other. By changing the applied voltage, the chemical potential can be changed to make it absorb more light in TM mode and less light in TE mode at a fixed value. After transmission over a distance, the entire device can be used as a TE polarization device by using different attenuation coefficients to eliminate the consumption of TM polarized light and preserve TE polarized light. The experimental results demonstrate that the extinction ratio of the TE-pass polarizer at the communication band (980 nm is up to 27 dB when the propagation distance is L ≈ 3.5 mm). At the same time, the offset has an exceedingly large bandwidth thanks to the fact that the properties of graphene do not vary with the operating frequency. It is worth noting that the chemical potential of graphene can be regulated, unlike conventional metal-clad fiber optic sensors that can only transmit TM mode. Apply voltage to the ferroelectric polymer with strong dipole moment coated on the surface of graphene, as shown in Figure 2b. When the chemical potential is greater than half of the photon energy, the polarizer converts from TE-pass polarization to TM-pass polarization [15]. Similarly, Zhang et al. [37] deposited graphene and a high refractive index (RI) index material Polyvinyl Butyral (PVB) on the side polishing fiber, in turn, in order to improve the coupling between graphene and evascular field, whose structure is shown in Figure 2c. PVB has a high RI to attract light. To increase the intensity of the interaction between light and graphene, the distance from the polishing region to the core is selected to be 1μm. Continuous waves at 1425~1600 nm are incident on the graphene fiber after passing through the polarizer, half wave plate, and microscope. The experimental results show that, when the wavelength of incident light is 1550 nm, the polarization extinction ratio is up to 37.5 dB and the transmission loss is 1 dB. In the range of 1425~1600 nm, the extinction ratio is greater than 26 dB. Unlike the polarizer structure that was designed by Zhang, Li [38] transfers graphene-coated 100-nm-thick gold to side-polished fibers. Additionally, the distance from the polishing zone of the device to the fiber core is 6 μm, which is slightly longer than the distance of the polarizer, where PVB is deposited. Apart from the above differences, the polarizer structures of the two are roughly the same. Graphene increases the evanescent field in the lateral abrasive zone when light waves pass through the abrasive zone. However, the losses of TM mode are relatively large since the metal surface can only generate TM mode plasma waves. The TE mode propagates in the optical fiber and graphene layer. The graphene-coated fiber does not significantly absorb the TE mode, and the polarization loss of the device is greatly increased. The polarizer can achieve an extinction ratio of 27 dB and insertion loss of 5 dB when the incident light wavelength is 1550 nm. However, when compared with the polished fiber, the micro-nano fiber has a stronger evanescent wave. He [39] coated graphene on the upper surface of the micro-nano fiber and placed it in the middle of the MgF_2_ (4cm(L) × 3cm(W)× 1cm(H)) base with low RI, as shown in Figure 2d. When the light is incident into the waveguide, more evanescent waves interact with graphene as the radius of the micro-nano fiber decreases. The simulation results show that the hybrid waveguide is suitable for TM-pass polarizer when the radius of the micro-nano fiber is less than 1μm. The extinction ratio is as high as 27 dB when the radius of the fiber is 0.8 μm and the propagation length is 3.5 mm. Graphene-microfiber hybrid waveguide has the advantages of easy fabrication and compatibility with optical fiber system. It has potential application prospect in pulse width adjustable pulse generation. In addition, micro-nano fiber has high sensitivity to the surrounding environment and it can be implemented to sensors in the future.

### 2.2. Graphene Hybrid Waveguide Polarizer

All photonic elements on the chip are required to be compatible with other functional planar waveguide devices in order to develop chips for next-generation photonic integrated circuits (PICs) that are compatible with complementary metal oxide semiconductor (CMOS). Fortunately, graphene is highly compatible with conventional CMOS devices and manufacturing processes [13,40,41]. Kim et al. [42] designed a hybrid waveguide polarizer based on a planar optical waveguide. The device is illustrated in Figure 3. Here, the graphene-based hybrid waveguide polarizer can be selectively used as a TE-pass or TM-pass polarizer by determining whether an upper cladding is present in the device. Upper and under cladding layers are both made of UV-curable perfluorinated acrylate polymer resin, which can adjust the conductivity and carrier density of graphene, so that graphene can selectively support TM or TE surface waves. The TE-pass polarizer is composed of graphene, a core with a rectangular cross section, and an undercladding, where the RI of the cladding and core is 1.37 and 1.39, respectively, as shown in Figure 3a. In addition, the waveguide is covered with air since there is no upper cladding. Adding a UV-curable polymer resin with the same RI as the under cladding layer to the top of the waveguide constitutes a TM-pass polarizer, which adjusts the electrical properties of graphene on the waveguide core, as shown in Figure 3b.

Kim first rotated a 20 μm-thick under-cladding onto a silicon wafer and then UV-cured it to fabricate the graphene-based polymer waveguide polarizer. The core material is subsequently distributed to the lower cladding to form a core 5 μm thick and cured with UV light. Graphene film that is grown using 300 nm thick nickel sputtering on a SiO_2_/Si substrate is mechanically transferred to the core by thermo chemical vapor deposition (CVD). The mask is first etched by using a photolithographic technique, and then the waveguide core is manufactured by modifying it with O_2_ plasma, in order to obtain a convex linear waveguide with a graphene tape as a core. In the experiment, the optical characteristics of the graphene polarizer based on the air-coated polymer were measured first, and the optical properties of the modified graphene polarizer were then measured again after the polymer resin was spin-coated to form the upper cladding layer. The output port of the waveguide polarizer that they made measured the infrared image according to the degree of polarization. The TE polarized light intensity of the air-clad graphene polarizer is stronger than the TM polarized light intensity. This implies that the key factor for TE-pass polarizers is the air-cladding waveguide. Conversely, the upper layer of the waveguide in the TM-pass polarizer is covered with a UV-curable polymer resin, which modifies the polarized light for a brighter spot. The intensity of TE polarized light is faintly visible. A slab mode can be observed since the RI difference between the upper and lower layers is exceedingly small. This means that the waveguide as a TM-pass polarizer. At the same time, they measured the insertion loss of the waveguide polarizer in order to further explore the characteristics of the conducting mode. Polarization has a significant impact on insertion loss when graphene strips are placed on the waveguide core. For air-coated graphene-based TE-pass polarizers, the average insertion loss of TM and TE polarized light was measured, which were 20.7 dB and 10.9 dB, respectively, and the extinction ratio is about 10 dB. When compared with the modified graphene-based TM-through polarizer, the measured TE insertion loss increased to 50 dB on average, and the TE insertion loss increased by 19.8 dB as compared to TM polarization. This shows that most of the optical power is radiated out of the channel waveguide in a slad mode. Thus, the modified graphene polarizer can be used as a TM-pass polarizer. Based on the experimental results, Kim concluded that the proposed graphene-based planar waveguide device can be further exploited for the development of on-chip PICs by taking extraordinary advantages of graphene s optical and electrical characteristics.

Pei [43] used poly (methyl methacrylate) (PMMA) film to reduce the doping level of graphene in contrast to Kim et al.’s use of UV-curable polymers to doping graphene to relatively high levels. He designed a simple broad band planar-lightwave-circuit (PLC) type polarizer based on PMMA-assiseted graphene/glass composite waveguide (PGGW). Graphene films are made on a copper substrate by the CVD method. After a series of operations, such as spin-coating a PMMA film on a graphene film, and removing a copper matrix by FeCl_3_ solution, the graphene film is mechanically transferred to the surface of the glass waveguide. Here, the glass waveguide has also undergone Ag+/Na+ heat exchange [44] and end polishing. Pei analyzed the attenuation constant of PGGW corresponding to different graphene chemical potentials μ_c_ in order to ensure the true reliability of the PGGM polarizer, and found that the performance of the polarizer will be improved, if μ_c_ decreases. The experiment proved that the maximum extinction ratio of the device at the wavelength of 1370 nm is approximately 28 dB, which is equivalent to the graphene/fiber polarizer.

### 2.3. Graphene THz Polarizer

THz technology was first generated in the end of the 19th century and researchers, such as Rubens, have proposed research in this band. In 1974, Fleming formally proposed the term “THz”, which was first used to describe the design of Michelson interference [45]. At that time, microwave theory and optical theory could not fully explain the wave band, and there was the absence of effective detection means and generation methods. Until the mid-1980s, scientists knew little about the nature of the band, known as “THz Gap” [46]. Since then, it has become a common technology to have a stable pump THz excitation light source with broadband, so THz technology began to develop rapidly due to the continuous development of new materials and the rapid development of new technologies (especially the development of ultra-fast laser technology). The response of graphene to THz is mainly an in-band transition, similar to the free-electron response of metal, which supports the propagation of SPP. Previously heavy metal such as gold and silver was considered the best surface plasmon materials. However, it obtains the disadvantage of high ohmic loss and uncontrollability. The most important characteristic of graphene as a SPP conducting material is its adjustable dynamic conductance. In the case of low doping, the virtual part of its dynamic conductance is negative, thus showing a semiconductor property, which can conduct TE surface waves. At high doping, the simulated part of the dynamic conductance is positive, which can conduct TM surface waves. At present, graphene THz devices mainly include: THz transmitter, THz modulator, THz isolator, THz polarizer, THz detector, and THz metamaterial, etc.

Polarization manipulation, as one of the most basic characteristics of electromagnetic wave manipulation, has received extensive attention. It has significantly promoted the development of many fields, such as optoelectronics, analytical chemistry, and biology. Traditional methods for realizing polarization operations include photoelastic modulators and optical gratings [47]. Although these methods can control polarization at will, they usually require large equipment and long propagation distances to achieve phase accumulation. Metamaterials, artificial subwavelength composites with unique electromagnetic responses, are widely used to manipulate the polarization states to alleviate this burden. Metamaterials exhibit remarkable responses to a wide range of desired frequencies. Due to these advantages, a large number of metamaterial based polarization manipulators are available for polarization conversion, such as linear to linear [48,49], linear to circle [50,51], and circle to circle [52,53,54]. For example, the chirality supersurface bonding method Fabry-Perot cavity resonance [55,56,57] was used to achieve broadband and multi-band asymmetric transmission with high transmission efficiency. Nevertheless, although these designs show high polarization conversion efficiency, once the pattern structure is fixed, their polarization conversion frequency band cannot be tuned, which hinders their wider application. On the other hand, tunable composite metamaterials are capable of dynamically manipulating THz waves while using external stimuli, such as heat [58,59,60], electrical offset [61], and photoexcitation [62,63,64,65], which open a bright prospect for designing general-purpose devices. For example, the optical excitation is used to realize the rotation conversion of chiral supramolecules and the electromagnetic control of the polarization of light, which has an important application prospect in the processing of THz waves [66]. With the unprecedented development of THz science and technology, the design of such tunable multifunctional devices is not only expected by people, but also the necessity to fill the THz gap.

In 2016, Yu et al. [67] proposed a wideband tunable polarization converter working in THz region reflection mode, which is realized by a hybrid metamaterial that consists of an I-shaped metal resonator and a double-layer graphene sheet. They found that, by changing the Fermi energy of graphene, the phase difference between the two orthogonal polarized components could be adjusted over a large range. In this way, the device can achieve electrical control of the Fermi energy of the graphene film without reengineering its structure to achieve linear-linear, linear-circular, and linear-elliptic polarization conversion of its functions. In the same year, Guo [68] proposed a novel wideband ultra-thin polarization converter, a reflective half-wave plate, as shown in Figure 4a. The half-wave plate consists of a group of L-shaped graphene sheets, which are placed on a dielectric spacer layer that is supported by a gold substrate to suppress transmission. He proved that the polarization converter angle was not sensitive by calculating the polarization conversion ratio (PCR) under different incident angles. Without changing the geometry of the graphene, their properties can be dynamically adjusted across the entire THz frequency spectrum by electrostatic gating the graphene. In 2018, Zhu et al. [69] proposed a new broadband sinusoidal slot graphene cross-polarization converter (CPC), which is sandwich structure with graphene sheet on the top, the ground on the bottom, and dielectric layer as spacer, as shown in Figure 4c,d. It can achieve broadband THz polarization conversion from 1.28 to 2.13 THz, PCR greater than 0.85. Based on the unique characteristics of graphene, the operating bandwidth, and size of PCR can be adjusted by adjusting the chemical potential and electron scattering time. Continuous plasmon resonances are excited at the edge of the gap by gradient-width modulation of graphene-based cell structures. Therefore, this device can realize broadband polarization conversion in a simpler structure when compared with other traditional broadband polarization converters utilizing multi-layer or multi-resonator structures. The same as Guo’s polarization converter, the polarization conversion properties of CPC are insensitive to incident angles. PCR remained above 0.85, even when the incidence angle increased 50°.

## 3. Polarization Sensors Based on Graphene

In recent years, graphene has been a great success in the field of biological and chemical sensing. Biochemical sensors that are based on graphene and its derivatives have been used to detect various small molecules, including gases molecules, small chemical ions, bacterial and viral cells, and dynamic cellular activity due to the photoelectric properties, biocompatibility of graphene materials, and the ease of biofunctionalization and high specific surface area of graphene derivatives. When compared with previous sensors, these sensors show superior performance in terms of sensitivity, resolution, detection range, and response time [70]. Next, this section mainly introduces its principles and applications from the perspective of biosensors and fiber optic sensors.

### 3.1. Graphene DNA Biosensor

Surface Plasmon Resonance (SPR) sensing technology is a new sensing technology that was developed in the 1990s. The SPR sensor developed by this technology is a high-resolution optical RI sensor that can be used to detect small changes in RI [71,72,73]. After years of development, SPR sensors have been widely used in the fields of chemical and biological detection, with the advantages of high sensitivity, low sample consumption, fast measurement speed, and no need for label processing. However, there are also disadvantages, such as noble metal film oxidation, high loss, and difficulty in combining with biomolecules. So far, researchers have tried various ways to enhance the sensitivity of the SPR sensor, including excitation of long-range surface plasmas [74], the use of resonant structures, such as metal nanoparticles [75,76] and optical gratings [77]. Graphene has excellent optical properties, large specific surface area, strong adsorption capacity, stable properties, and chemical crosslinking. Many scholars have applied it to SPR sensors to improve the sensitivity of SPR sensors, reduce metal film oxidation, and achieve specific detection. The theoretical research of Fu et al. [78] showed that graphene coating on the precious metal layer of the SPR sensor could effectively increase the sensitivity of the sensor. In 2012, Salihoglu et al. researched the surface plasmons that were produced by graphene-modified metal surfaces. Graphene was transferred to the surface of the metal film for the first time to stimulate the SPR phenomenon on the surface of the metal film that was modified by graphene and study the non-specific binding reaction between the protein molecule and graphene [79].

In 2019, Sun et al. designed a highly sensitive polarimetric controlled-modulation plasma biosensor based on monolayer graphene/gold nanoparticles for the direct detection of single-stranded DNA (ssDNA) hybridization [80]. The system is based on monolayer graphene/gold-NPs hybrid sandwich architecture. The composite structure of two-dimensional graphene and Au nanoparticles (Au-NP) generates electromagnetic oscillations that formed by the interaction of free electrons and photons on the surface of the Au film. This causes a strong plasmon field enhancement effect, which results in a significant increase in sensitivity to the detection target. By optimizing the positive/negative charge combination method of gold nanoparticles and ssDNA, SPR sensors can be monitored in real time, and the entire process becomes more simple and quantifiable. Graphene is a single-layer graphene sheet that is prepared by a low-pressure chemical vapor deposition (LPCVD) method, which is uniform and consistent without complex surface functional groups. In addition, the detection limit of the target ssDNA is one to three orders of magnitude higher than the GO and reduced graphene oxide (rGO) sensors that have been previously studied [81,82].

The SPR sensor that is based on polarization adopts the coupling structure of Kretschmann prism. The SPR system is shown in Figure 5. Sun mentioned that different graphene layers on the surface of the gold film have different sensitivities in polarization-controlled modulation. The single-layer graphene structure has the highest sensitivity response, being close to 1.9 times of the sensitivity of bare gold membrane structure. The excited plasmons on the surface of the linene are sensitive to the changes in the effective permittivity of the surrounding environment. Excited plasmons on the surface of graphene are sensitive to changes in the effective permittivity of the surrounding environment. They prepared a graphene/gold monolayer sensor chip while using LPCVD as part of calibration to verify the consistency of theoretical and experimental results. NaCl solutions of different concentrations (0.1%, 0.2%, 0.5%, 2%) are injected into the sensor through the syringe pump for a certain period of time. The sensor is injected with deionized water to obtain a series of measurements. The NaCl solution is then recirculated and tested at different concentrations. They compared the reflectance response of RI changes of the monolayer graphene/Au films and pure Au films calculated while using different concentrations of NaCl solutions. According to the experimental data, the sensitivity of single-layer graphene is almost twice that of bare gold (1.23 × 10^5^ pixel/RIU). The experimental results are consistent with the simulation results. Gold nanoparticles are also used as magnifying tags to enhance SPR sensing signals. The coupling between the local SPR of the gold nanoparticles and the propagation-SPR of the gold surface amplifies the SPR response, thus resulting in a significant field enhancement effect. They performed simulations on gold nanoparticles with a diameter of 10 ~ 40 nm to optimize gold nanoparticles. It was found that gold NP exhibited the best response at a diameter of 30 nm. This phenomenon reflected that when 30 nm Au-NPs are coupled with a graphene-coated Au film, the structure shows the highest absorption efficiency and the lowest scattering efficiency [83]. Moreover, the linear dynamic range of 10^-15^ M to 10^-7^ M of the sensor and the detection limit of 500 aM of the target ssDNA are also shown through DNA hybridization experiments. Sun believes that, when compared with previous DNA detection methods, gold nanoparticle-based biosensors have better application prospects.

### 3.2. Graphene Cell Biosensor

The research on graphene polarization selective absorption is based on the development of prismatic SPR sensors. In recent years, Xing et al. [84,85] found that removing the metal layer in the Kretschmann model and directly replacing it with graphene could also be used for surface sensing, which opened up a new direction for the optical sensing of graphene. Through theoretical analysis and experimental research, they found that, under total reflection, graphene’s absorption of s-polarized light is higher than that of p-polarized light [85], which is the opposite of the SPR phenomenon. When SPR occurs, p-polarized light is strongly absorbed, while s-polarized light is basically unchanged. In addition, Xing et al. conducted experiments on single-layer, double-layer, and four-layer graphene in the band range from 420 nm to 750 nm, and verified the theoretical analysis results. It is found that, as the number of graphene layers (one to four layers) increases, the reflectance of TE mode light gradually decreases, while the reflectance ratio (P / S) gradually increases. Therefore, while using four-layer of graphene as the sensing material, the performance of the polarization sensor is better than that of the single-layer and double-layer of graphene. It is concluded that, under total reflection, graphene absorbs more s-polarized light than p-polarized light. In turn, they used this property to explore the potential of graphene in information storage [86] and achieved an accurate determination of graphene layers [87].

In the field of sensing, Xing et al. designed a graphene-based optical RI sensor that can accurately detect a small number of cancer cells in normal cells at the single-cell level [84]. RI has long been considered as an important biophysical property in many fields, including cell growth monitoring, viral detection, and cancer diagnosis. Substantial evidence proves that the cell’s RI is associated with diseases, such as cancer, malaria, anemia, and bacterial infections. Optical measurement of cell RI is non-invasive and highly sensitive. Therefore, it has been widely used to determine or characterize biophysical properties without causing cell damage. The sensor used high temperature reduced graphene oxide (h-rGO), which was obtained by controlling the thickness of graphene. Its sensitivity can reach 4.3 mV/RIU and the resolution up to $1.7\times 10^{-8}$. In addition, a wide dynamic range can be generated by adjusting the incident angle and incident power. Single-layer and multi-layer CVD graphene has the obvious advantages of low defect, uniformity, and clear structure. However, the optimal thickness that is calculated by the standard optical constant of graphene (n = 2.6 + 1.3i) is 20 layers in order to achieve the most sensitive sensing, so the CVD method is difficult to grow. In addition, if CVD graphene is used, damage, peeling, pollution, and rough surface of folds will inevitably occur during the transfer process. It also causes the polarization dependence absorption effect to decrease. Therefore, h-rGO is the best choice compared to other types of graphene. In this research, they used an improved RI sensing model to insert graphene layers between high RI medium 1 and low RI medium 2 (Figure 6a.). Medium 1 and 2 correspond to $n_{1}$ and $n_{2}$. Figure 6b shows angle-dependent reflectance ratio diagrams for different h-rGO thicknesses. The solid line represents the calculation result and the solid point represents the experimental result. When the incidence angle θ_1_ approaches the critical angle θ_2_, the resolution and sensitivity can be improved by using the best thickness of graphene.

The resolution and sensitivity, which are the important parameters of RI sensing, are calculated with the following formula:$$S=\frac{{\left.\frac{d\Delta R}{dn}\right|}_{n=n_{2}}•p_{0}}{a}\;L=\frac{V_{n}}{S}$$

Among them, S and L represent the sensitivity and resolution of RI sensing, respectively. α is the response of the balanced detector, $P_{0}\;$is the incident power in either TE or TM mode, and $V_{n}$ is the value of the noise signal. In the formula above, some parameters are known quantities. Here, the thickness of h-rGO is 8.1nm and its optical constant is $\hat{n}$ = 2.6 + 1.25i. The incident angle is approximately equal to the critical incident angle, and the power is fixed at 80 μW. α is 0.00361 μW/mV and $V_{n}$ is approximately 10 mV. The h-rGO enhanced RI sensor is used to detect ultra-low concentration sodium chloride solutions of different concentrations in real time, as shown in Figure 6c. It can be seen that the slight but significant difference between the RI of 0.003% sodium chloride solution (n = $n_{water}$ + 0.000144) and the RI of water is about 210 mV.

Figure 6d presents the time-dependent changes in voltage that correspond to mixed 5 and 6 μm standard polystyrene (PS) microspheres as they roll through the detection window. This shows that the graphene-based optical single-cell sensor (GSOCS) is extremely sensitive to the monitoring of single cells (like PS samples). In the disposable detection diagram, the light blue and light purple regions represent discrete voltage signals of 5 and 6 μm microspheres, respectively. At a high speed of 3μL/hour, the reference voltage signal remains relatively constant at -5 ± 0.01V, so it is concluded that the h-rGO sensor can ensure accurate repeat detections without surface contamination. Figure 6e shows the real-time sensing process of sensing a single 6 μm PS microsphere by GSOCS, including the size of the period and the corresponding voltage change. The change in voltage signals at different locations on a single microsphere and the number of microfluidic flows can be detected because of the surface uniformity of the microsphere. Figure 6f shows the discrete change in voltage of a single lymphocyte or Jurkat cell as it rolls in the microfluid over time. The actual photos of I-II-III in Figure 6g correspond to the I-II-III region in Figure 6f. Here, the size and number of individual PS microspheres are factors that affect the time variation (Δt) of the PS microspheres rolling on the detection window. The Δt of a single PS microsphere is calculated to be approximately 28 ms, and it shows that the voltage signal can respond quickly in tens of milliseconds.

In summary, this highly sensitive graphene optical sensor enables unlabeled, living, and highly accurate detection of a small number of cancer cells in normal cells at the single-cell level, as well as simultaneous detection and differentiation of two cell lines without separation. It provides an accurate statistical distribution of normal cells and cancer cells with fewer cells. The hypersensitivity and resolution of GSOCS to RI play an important role in promoting the dynamic study of individual cells in the cellular submicro structure. In addition, the high resolution and sensitivity of sensor to RI measurements can be extended to other areas, such as drug discovery, environmental monitoring, and gas- and liquid-phase chemical sensing [84]. The application of graphene in cell sensing detection has a profound impact on the field of health care. Early detection is the best way to prevent the spread and deterioration of malignant cells, such as cancer cells. The excellent sensing performance of graphene makes the test results more reliable. It is believed that it can play an important role in the future anti-cancer health care business.

### 3.3. Graphene Optical Fiber Sensor

As an important branch of sensors, optical fiber sensors integrate signal acquisition and transmission. In addition to the excellent physical properties of the optical fiber itself, its functional characteristics have gradually become apparent with the introduction of new functional materials, such as graphene. Over the past three decades, optic fiber sensors have been used in many fields, such as energy, environment, and biomedicine, due to its unique ability to support biocompatibility. Optical fiber has the properties of anti-electromagnetic interference and atomic radiation, fine diameter, soft quality, light weight mechanical properties, insulation, non-inductive electrical properties, water resistance, high temperature resistance, corrosion resistance chemical properties, and many other excellent properties [88,89]. Moreover, with the maturity of photonic crystal fiber technology, the diversity and flexibility of its structure make Fiber have broader development space in the field of sensors.

Fiber optic sensors can be divided into two types according to the sensing principle: one is the light-transmitting sensor and the other is the sensing sensor. In the light-transmitting optical fiber sensor, the optical fiber is only used as a carrier of light and there is no special requirement for the structure of the optical fiber itself. The sensor depends on the direct action of the optical signal to be measured and then calculates the change to be measured by observing the spectral change of the optical signal. In the optical fiber sensor, the optical fiber is only the carrier of the light, and there is no special requirement for the structure of the optical fiber itself. The sensor depends on the direct action of the optical signal to be measured and it calculates the change to be measured by observing the spectral change of the optical signal. This type of sensor is simple in structure, but the physical quantities that can be measured are limited and the sensitivity is low. In the sensing fiber sensor, the fiber is both the main body of the sensor and the medium of signal transmission. Changing the structure of the fiber to increase the type of sensing and sensitivity of the sensor. For example, pulling cone, grating, polishing, and corrosion process the sensing part of the optical fiber. With the development of the diversity of optical fiber structures and the improvement of the precision of sensing detection, graphene-based optical fiber sensors have been developed for research and application. Various graphene fiber optic sensors show significant performance advantages when compared with traditional fiber optic sensors.

In 2014, Wu et al. [90], for the first time, laid a layer of graphene under micro fibers to make a hybrid optical waveguide. The TE mode is significantly attenuated when the acetone molecule is adsorbed to the surface of graphene. The acetone sensor has excellent sensitivity and repeatability. Additionally, the structure is so small that it is easy to integrate and encapsulate (~2 μm, whereas photonic crystal fiber gas sensors and F-P gas sensors typically have probe diameters of 60~200 μm). Therefore, the fiber optic sensor belongs to sensing fiber optic sensor. The schematic diagram of the gas sensor based on a hybrid graphene-waveguide coupled with a silica microfiber (GWMF) GWMF structure is shown in Figure 8a. Microfiber, as the optical carrier, provides a large proportion of attenuation field outside the fiber and it plays a key role in optical coupling and acquisition in graphene waveguide (GW). Through van der Waals and electrostatic attraction, microfiber and graphene films can be combined well. Transmitted light is introduced from the input single-mode fiber (SMF) and coupled through a fiber taper. When the transmitted light is coupled to the graphene layer, it can partially penetrate the graphene thin film and propagate along the graphene thin film by evanescent wave generation with GW. At the output port, the light can be collected by another tapered fiber that is derived from the multi-mode fiber (MMF). In this configuration, they used SMF emission fibers to ensure that the$\;HE_{11}$ mode was mainly in the input light and collect the transmitted light from GW while using MMF, which could preserve the high order polarization modes that are generated by GW. This is important for detecting polarization patterns in gas sensing. The $HE_{11}\;$mode emitted to GW is evanescent wave, as shown in Figure 7a. The electric field of evanescent wave includes x polarization component (TM) and y polarization component (TE). The transmission intensity of TE mode along GW is much higher than that of TM mode [26]. As shown in Figure 7b, van der Waals forces change the initial hexagonal structure of graphene when the gas molecules are distributed over GW. As the internal atoms of carbon attract or push each other to change the spatial distribution of carriers [91], this phenomenon will change the lattice of graphene crystals. As a result, the local dielectric constant of graphene changes. While considering relationship$\;\omega^{2}\mu_{0}\varepsilon =\omega^{2}\mu_{0}\varepsilon_{r}=\varepsilon_{r}k_{0}^{2}=n_{eff}^{2}$=$k^{2}$, effective RI $n_{eff}$ varies with permittivity. After theoretical formula derivation, the attenuation coefficient of TE polarization mode could be expressed as $\omega n_{eff}^{IM}\times$^z^/_c_. As a lager negative imaginary part of $n_{eff}$ causes a larger attenuation, gas molecules accessing would increase $n_{eff}^{IM}$, and the transmission light attenuation can be observed. The experimental results show that the sensor has a sensitivity of 0.31 dB/100 ppm and excellent repeatability, which can be widely used in biology, chemistry, medicine, and other fields.

The team further proposed a graphene multimode fiber interferometer (GMMI) In order to improve the problem of light leakage [92]. The numerical simulations and experimental results show that graphene can effectively enhance the evanescent field strength of the surface of microfibers, thereby increasing the Sensor sensitivity. The lower detection limits for ammonia and water vapor are 0.1 ppm and 0.2 ppm, respectively. Figure 7c shows the schematic of GMMI. The graphene monolayer is wrapped in microfibers that are tightly attached to the MgF_2_ substrate. The SMF with a core diameter of 8 μm was chemically etched to obtain ultra-fine optical fibers with a diameter of d = ~ 10 μm. Among them, the length of the entire fiber tapered portion is approximately ~ 3 cm. Here, the middle of the tapered portion is selected to be coated with graphene, and its length LG is ~ 3 mm. At a wavelength of 1550 nm, the RI of the MgF_2_ substrate, the cladding of the SMF, and the SMF core were measured, which were 1.37, 1.44, and 1.45, respectively. This research breaks through the application of graphene-based planar waveguides and extends to cylindrical waveguides in the optical field, laying the foundation for the development of fiber devices that are based on graphene polarization characteristics.

## 4. Polarization Modulator Based on Graphene

As an important part of the optical system, optical modulators are devices used to regulate the basic characteristics of spatial light or waveguide light transmission that is controlled by electrical signals. How to further improve the bandwidth, reduce the half-wave voltage, and improve the modulation efficiency. Meanwhile, on this basis, how further reduce the modulator’s size and interference by external factors has been the research focus of modulator in recent years. When compared with traditional modulators, graphene modulators have the following advantages. First, graphene presents a strong intra-band and inter-band light transition, which has a strong coupling with light, resulting in higher modulation efficiency. Second, graphene has an excellent fast carrier movement rate at room temperature up to 200,000${\mathrm{cm}}^{2}V^{-1}S^{-1}$[93,94], and the corresponding resistivity is only $10^{-6}$Ω. Therefore, the carrier can realize the fast change of Fermi energy level through the band filling effect, so that it can be modulated at high speed. Third, graphene’s absorption of light in the communication band and the far-infrared region is independent of the wavelength, which makes the graphene modulator have an exceedingly large bandwidth [95]. In addition, the interaction area between graphene and the light field can be increased by increasing the number of graphene layers to obtain better modulation characteristics. In 2011, Liu and X.B. Yin et al. prepared graphene modulators for the first time, which is a major breakthrough in the application of graphene to new modulators. In the paper [41], he pointed out that the research on integrated optical modulators has reached a bottleneck. It is difficult to reduce the size of the modulator below the μm level due to the low electro-optic effect of conventional semiconductor silicon-based modulators. Meanwhile, the modulators of other semiconductor materials do not mesh well with silicon-based optoelectronic integration platforms. At present, finding a light modulator material that is compatible with the CMOS platform and has high-speed modulation efficiency has become a new focus of researchers. The experimental verification of the new material graphene no doubt gives scientists hope. It has an ultra-low resistivity that allows for electrons to run at ultra-high speed during it, and is the smallest resistivity material in the world today. At the same time, graphene has zero band gap and unique light absorption, which makes it exceedingly suitable for making light modulators [41].

### 4.1. Waveguide Graphene Modulator

At present, further miniaturization limits silicon-based optical devices, which makes it difficult to meet the development requirements of high speed and high broadband. Like silicon-based light modulators, Si crysta does not have linear electro-optic effect, and the secondary electro-optic effect is weak, thus limiting the modulation bandwidth and extinction ratio of the device. Therefore, finding a material with excellent stability and reliability, which is compatible with CMOS technology, and sufficient modulation bandwidth and modulation efficiency has become an important research content of scientists. The emergence of graphene has brought new challenges. It has the characteristics of zero-bandgap band structure and exceedingly high carrier mobility, thus making it a promising application in the field of optical modulation.

The unit cell as an absorber, like a graphene double-layered band, only absorbs electromagnetic waves of a specific polarization without affecting cross-polarized waves. Zhang proposed a polarization modulation scheme through electromagnetic wave reflection to further study this characteristic [96]. The polarization modulator is a combination of two polarization-related metamaterial absorber structures, and its cell is composed of graphene strip, thin layer of silicon dioxide, gold cut-wire, and polymer substrate, which are stacked and combined several times in sequence. Among them, two orthogonal gold cut-wires change the incident electromagnetic wave. Additionally, the amplitude and phase of the reflection can be calculated to reflect the modulator’s response to different polarized waves. The results show that the two orthogonal polarized waves reflect nearly the same amplitude at different Fermi energies, but the reflection phase is slightly different. Therefore, the bias voltage can be adjusted to control the reflection coefficient of x- or y-polarized waves, respectively. Thus, the polarization state of the reflected wave can be flexibly modulated. Based on the reflection amplitude and phase results calculated above, Zhang calculated the azimuth angle θ and axial ratio (AR) of the reflected waves at different bias voltages under normal linear polarization incident light illumination at 45°. The experimental data display that the polarization azimuth of the reflected wave at 0.87 THz can be controlled by the bias voltage, which continuously changes from 0° to 90°. Above 0.87 THz, the modulation range of the azimuth angle comes to reduce from the full range of 0° to 90°. The reflected waves become elliptically polarized by the superposition of the two orthogonal components to further study this characteristic. 

The above device adopts the combination of a straight waveguide substrate and graphene, and then adjusts the absorption coefficient of graphene through an electric field to realize the modulation of the device. One of the problems with such structures is that that the interaction between graphene and light field is so weak that the modulation depth is low. The buried waveguide and slit waveguide can theoretically solve this problem. In 2017, Yang et al. [97] produced a polarization modulator based on a graphene and silicon waveguide hybrid structure, which has the advantages of small size, high extinction ratio and compatibility with CMOS process, as shown in Figure 8a. The device enables the operation and selection of polarization due to changes in the effective RI of graphene when applied voltage is applied. Since previous studies have found that epitaxial multilayer graphene is consistent with single-layer graphene in such characteristics as high carrier mobility and infrared transmission [98,99,100], they used the optical constants of single-layer graphene to calculate the multi-layer sandwich structure of the device. In the white area of the Figure 8a, five layers of graphene are embedded in the middle of the silicon ridge. The height of each Si_3_N_4_ layer is d = 10 nm, and n_Si3N4_ =1.98, represented by the green area in the Figure 8a. The chemical potential plus the adjustable voltage causes a change in the RI, which determines whether the TM mode is excited in the graphene waveguide. However, if the design parameters are changed, including width and height, the TE and TM modes are excited and transmitted in the hybrid waveguide. Additionally, while using a sandwich structure, Ye et al. embedded two layers of graphene and a buffer layer in a silicon ridge waveguide, as shown in Figure 8b, to enhance the interaction between graphene and light [101]. By regulating the graphene layer voltage, the TE mode effective RI can produce a considerable range of variation, which is suitable for use as an optical phase modulator. When the device length is 75.6 μm, π phase conversion can be realized, the modulation bandwidth can reach 119.5 GHz, and the loss is as low as 0.452 pJ/bit.

### 4.2. Mach-Zehnder(M-Z) Graphene Modulator

As a mature electro-optic modulator, M-Z modulator is widely used in analog photonic link. The M-Z modulator is different from the electrical absorption modulator, which depends on the applied voltage to change the absorption spectrum of the material. It mainly uses the linear electro-optic effect to adjust the RI of the material, and then uses the M-Z interferometer structure to make the output optical power change with the applied voltage. In practical applications, LiNbO_3_ is one of the most commonly used materials for making M-Z modulators. The effective RI of LiNbO_3_ waveguide can be changed by adjusting the electric field intensity at both ends of LiNbO_3_ waveguide when used to make the M-Z modulator because of the first-order electro-optic effect of LiNbO_3_ material, thus changing the phase difference between the upper and lower arms of the modulator. Subsequently, the intensity of the output light is changed by the interference of the upper and lower arms, and the process of loading the electrical signal into the optical signal is completed. However, the size of the M-Z modulator that is based on LiNbO_3_ is generally larger, so the traveling wave electrode structure is needed to improve the modulation rate. In addition, the biggest limitation of the LiNbO_3_ M-Z modulator is that it is difficult to be compatible with existing CMOS processes, which makes integration applications difficult. Thus, people focus their attention on the most commonly used integrated electro-optic device material—silicon. The early silicon-based M-Z modulators were all based on the thermal and optical effect due to the high thermal and optical coefficient of silicon materials. However, such devices are limited by the speed of thermal modulation, so it is difficult to meet the demand of high-speed signal modulation. The modulation rate of the fastest thermosensitive silicon-based modulator published so far is still on the order of microseconds [102]. Subsequently, it is found that the free carrier dispersion effect in monocrystallone silicon led to greater variation in waveguide RI and absorption coefficient. Although the free carrier dispersion effect of monocrystallone silicon can improve the RI and absorption coefficient of waveguide, the improvement is still limited. With the continuous research on graphene materials, it is found that the combination of graphene and silicon-based waveguides can greatly improve the variation of the effective RI and absorption coefficient of waveguides. Moreover, the M-Z modulator changes the RI of graphene on the phase shifter, and ultimately uses interference to control the output light intensity, unlike the electro-absorption modulator, which changes the absorption coefficient of graphene to control the output light intensity. It avoids the disadvantage that the absorption rate of single-layer graphene is only about 2.3%, thus greatly increasing the extinction ratio of the modulator. In 2012, Grigirenko et al. [103] proposed the first graphene-based M-Z structure light modulator. The superposition of the silicon base and the graphene layer was used as one of the interference arms of the M-Z structure, which greatly enhanced the electrical refraction efficiency of the arm. 

In 2013, Yang et al. designed an M-Z modulator that was based on a hybrid graphene silicon waveguide (GSW) that was ultra-fast and compatible with CMOS [104]. As shown in Figure 9a, GSW is made on a rectangular silica substrate, and its two arms are separated from each other at a fixed angle. Here, a lateral slot waveguide is formed by interposing Si_3_N_4_ inside the silicon. Three layers of graphene sheets were placed in Si_3_N_4_, and they were separated from each other by 10 nm, like a sandwich. Three-layer graphene was chosen over single-layer graphene, because the effective index variation of the GSW can be larger. The metal electrode contacts the graphene sheet to form a capacitive structure between the second graphene layer and the first and third graphene layers. The slot waveguides have played a crucial role in signal processing in some previous studies [105,106]. The graphene sheet placed in Si_3_N_4_ has a greater effect on the GSW effective index ($n_{eff}$) than the graphene sheet placed on top of the silicon waveguide, owing to most light energy is confined within the gap. Yang experimentally analyzed the changes in the dielectric constant of graphene and the $n_{eff\;}$of GSW in TM mode. When compared with the $n_{eff}\;$provided by the traditional material Kerr effect, the TM mode in the GSW has a larger $n_{eff}$ change, that is, it has a strong phase modulation capability. Additionally, they found that when the dielectric constant of graphene is close to zero, its μ_c_ is about 0.5 eV, which means that the epsilon near zero (ENZ) point. In addition, the GSW loss and the real and imaginary parts of $n_{eff}$ change significantly around the ENZ point [104]. For M-Z modulators, the phase of light on both arms can be uniformly moved in different directions simultaneously. At the same time, with the change of light phase, the loss difference between the two arms is still small. The phase change between the output signal and the input signal can be eliminated by using this characteristic. In addition, they also studied the effect of temperature on GSW $n_{eff}$. The experiments have shown that the modulator has excellent thermal stability at room temperature, but better performance at lower temperatures. When the environment becomes cold, GSW can obtain a large phase modulation capability, while the loss is still low. This can significantly reduce the insertion loss of the modulator. In addition, it is possible for this TM modulator to operate at ultra-high speed for ultra-high carrier mobility in graphene.

In 2015, Hao et al. proposed a polarization modulator that is based on graphene-coated interferometers that is independent of the polarization of incident light [107]. By applying different bias voltages on the graphene layer, it is possible to achieve different changes in the effective index ($n_{eff}$) of the TE polarization and TM polarization modes. The M-Z interferometer can be used to adjust the light transmission. The new device can be used as either a TE polarizer or a TM polarizer, depending on the external voltage applied to the graphene. Figure 9c shows the schematic of the modulator. The entire device is constructed on a silicon dioxide substrate, and the M-Z interferometer is made of a silicon waveguide coated with a graphene layer. The cross section of a silicon waveguide is a square with a side length of 300 nm. They first transferred the underlying graphene layer to the silica surface [40], and then placed a 7 nm transition layer alumina on the underlying graphene layer [41,108]. The process of directional growth or bonding techniques constructed the silicon waveguide, and the top alumina transition layer and graphene can be placed on the top surface of the doped silicon waveguide. Finally, in the same way, transitional alumina and graphene layers can be deposited on the sides of the waveguide. In addition, the graphene layers are separated from each other and extended to connect with the electrodes to provide an external oxide of graphene. Its extinction ratio is up to 19.15 dB in TE mode and 20.68 dB in the TM mode.

## 5. Polarization Photodetector Based on Graphene

As a kind of photoelectric device sensitive to incident light polarization, polarization-sensitive photodetector has a strong influence in navigation, astronomy [109], quality evaluation of mechanical industry [110], and many other scientific and technological fields. Most of the photodetectors use the photoelectric effect inside the material, in which the absorption of photons causes the carrier to be excited from the valence band to the conduction band, and then the output current. However, its spectral bandwidth is usually limited by the absorption bandwidth of the material. For example, photodetectors that are based on IV and III-V semiconductors are limited by the long wavelength limit. Graphene has a broadband property for light absorption. It can absorb light from terahertz to ultraviolet light. This wideband absorption property allows for graphene to be used as a photodetector for detection over a wider range of light. In 2009, Xia et al. Used single-layer graphene obtained by mechanical peeling as the active layer of the device, and selected Ti/Pd/Au metal materials as the source and electrode of the device. SiO_2_/Si was selected as the back gate substrate with Si as the back gate electrode. The first photo graphene photodetector was prepared to realize the super-speed photodetection of graphene [111].

Long wave infrared (LWIR) photodetectors have important applications in optical communication, night vision, and many other fields. Polarization is one of the most important properties of light, so polarization sensitive photodetectors have important value in practical LWIR applications. In recent years, various materials with special structures have been proposed to detect polarized light from ultraviolet to infrared light [73,112]. In 2019, Wu reported a self-powered photodetector with excellent performance. The detector consisted of graphene/PdSe_2_/germanium heterojunction, and enhanced light absorption of the van der Waals heterojunction of mixed size and effective carrier collection of graphene transparent electrode, which exhibited high polarization sensitivity and broadband advantages, including high light responsiveness, high specific detection rateand fast response speed, etc [113]. Wu noted that the polarization sensitivity of photodetector is extremely high, at 112.2, which represents the best result for two-dimensional (2D) layered material-based photodetectors. Many two-dimensional materials, including transition metal dihalides (TMDCs, such as ReS_2_, GeSe_2_, and ReSe_2_), black phosphorus (BP), and GeAs, exhibit excellent electronic and photoelectrical properties. However, they all have their own drawbacks. For example, the band gap of many TMDCs is usually greater than 1 eV, which is not suitable for infrared detection [114]. BP is not compatible with large-scale manufacturing processes and it is relatively unstable in the environment [115,116]. Therefore, Liu et al. proposed a new structure that was based on graphene nanoribbons for the design of polarization sensitive photodetectors working outside long red waves [117]. Figure 10 shows the structure. The graphene polarization sensitive photodetector (GPSP) consists of cross-sized graphene nanostructures that are deposited on a silicon dioxide substrate. Polarized light is incident perpendicular to the plane from top to bottom. Vertical and horizontal local surface plasma excited states both interact due to the presence of two graphene bands. They used a finite-difference time-domain method to obtain transmission spectra with polarization angles of 0° to 90° at intervals of 10°. While using the external wavelength to adjust the energy band of graphene and changing the absorption peak of local surface plasmons, the absorption wavelength of the device can be easily adjusted in the infrared range of 10.5 μm to 16.5 μm. Moreover, the corresponding photoresponsivities was studied in depth under the premise of polarization of incident light at a resonance wavelength of 12.58 μm. Under the polarization of incident light at 0° and 90°, the polarization-dependent photoresponse reaches the maximum and minimum, which are 1.717 A/W and 0.212 A/W, respectively. Simultaneously, the polarization sensitivity increases first and then decreases when the polarization of incident light is from 0 ° to 90 °. The results show that the designed photodetector exhibits polarization sensitivity. 

It is still difficult to obtain a graphene photodetector with both high photoelectrical responsiveness and fast response speed, according to the current research situation. It is necessary to consider the normalized geochemical measurement related to noise, photosensitive area and measurement bandwidth. Moreover, while improving the photoelectric responsiveness of the graphene photodetector, it might also reduce its ability to detect wide spectrum. Therefore, the research on improving the performance of graphene photodetectors should be carried out according to the specific application direction.

## 6. Conclusions and Future Works

As a new two-dimensional material, graphene has excellent optical and photoelectrical properties. Its spectral response range is extremely wide, achieving full spectral response between ultraviolet and THz bands. Graphene is an ideal optoelectronic material due to its high carrier mobility and ultra-high light response speed [118]. The development space of graphene polarizers is huge. In theory, graphene has a potential bandwidth of 500 GHz. The current research focus is to use new structures, new principles, and new methods to improve the sensitivity, responsivity, and compatibility of polarizing devices. Graphene and light interact differently in various photonic and optoelectronic devices, so graphene optical devices need to work under different structures [119]. The polarizing devices that are mentioned in this letter all require strong interaction between graphene and light. The single-layer graphene only absorbs 2.3% of the incident light in space. Therefore, finding a way to further enhance the effects of light and graphene is a new challenge. In addition, the shape, size, number of layers, electron bandgap structure, and purity of the graphene grown in the experiment are all uncertain. These factors will affect the conductivity and other characteristics of the sample, thus affecting the performance of graphene-based polarization devices. The preparation, transfer, repeatability, and compatibility of graphene with CMOS processes are all problems to be solved. However, graphene will have great application prospects, with the continuous improvement of the graphene preparation process and the continuous reduction of costs. It is especially excellent optical properties, graphene materials are expected to further promote the research and application of new polarizing devices based on the extraordinary optical, electrical, mechanical, thermal, and mechanical properties of graphene.

## Figures and Tables

**Figure 1 ijms-21-01608-f001:**
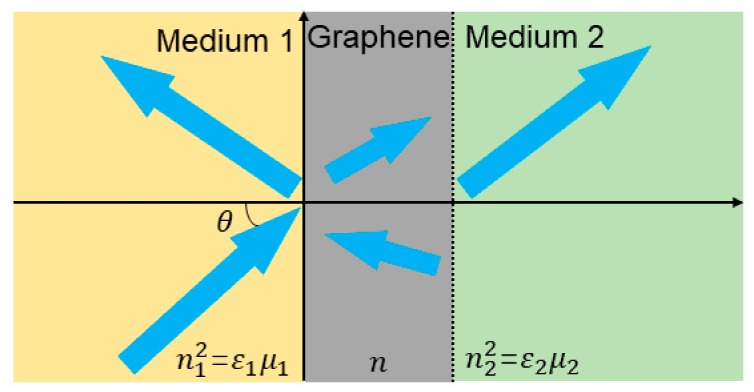
Schematic of a graphene layer sandwiched between two dielectrics (RI, n1 > n2).

**Figure 2 ijms-21-01608-f002:**
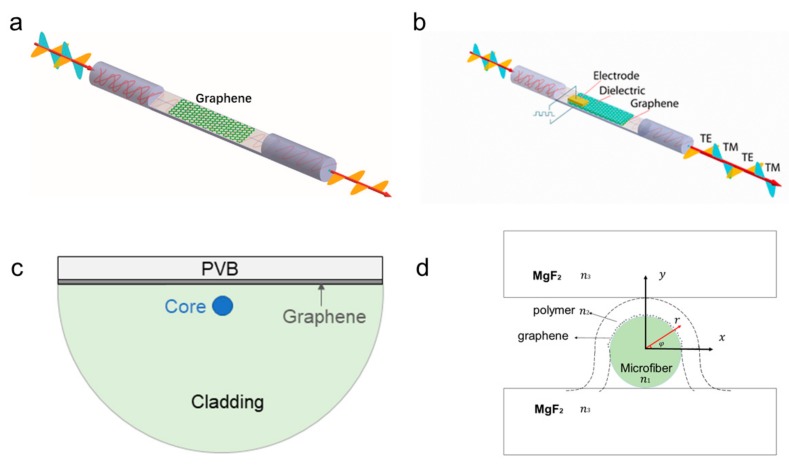
(**a**) Schematic illustration of the transverse electrical (TE)-pass polarizer. (**b**) TE-pass and transverse magnetic (TM)-pass switchable polarizers. (**c**) Cross-sectional view of polarizer structure, including Polyvinyl Butyral (PVB), graphene, and core. (**d**) Schematic of the cross section of the graphene-microfiber hybrid wave guide, in which polymer-supported graphene films cover the upper surface of the microfiber, sandwiched between two low-index substrates of the MgF_2_ (RI, n_1_ = 1.4443, n_2_ = 1.367, n_3_ = 1.37). Reproduced from [15] with permission of the American Chemical Society.

**Figure 3 ijms-21-01608-f003:**
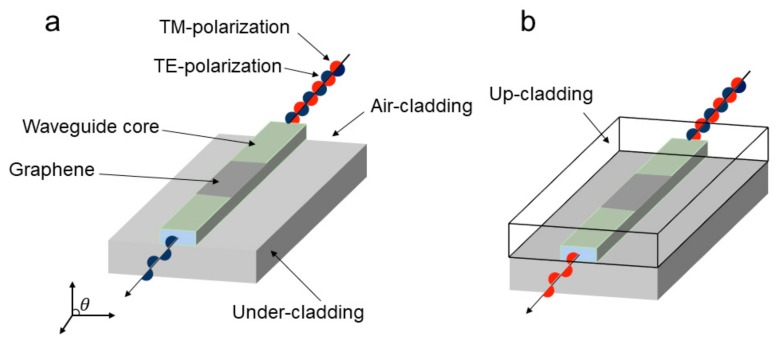
Schematic of a polymer-based planar-lightwave-circuit (PLC)-type graphene polarizer. (**a**) The polarizer serves as a TE-pass polarizer because the graphene strip with the air cladding supports TE mode surface wave. (**b**) Due to the fact that the waveguide is covered with a UV-curable polymer resin, the electrical properties of the graphene strip are adjusted to support the TM-mode surface wave. Thus, the waveguide serves as a TM-pass polarizer.

**Figure 4 ijms-21-01608-f004:**
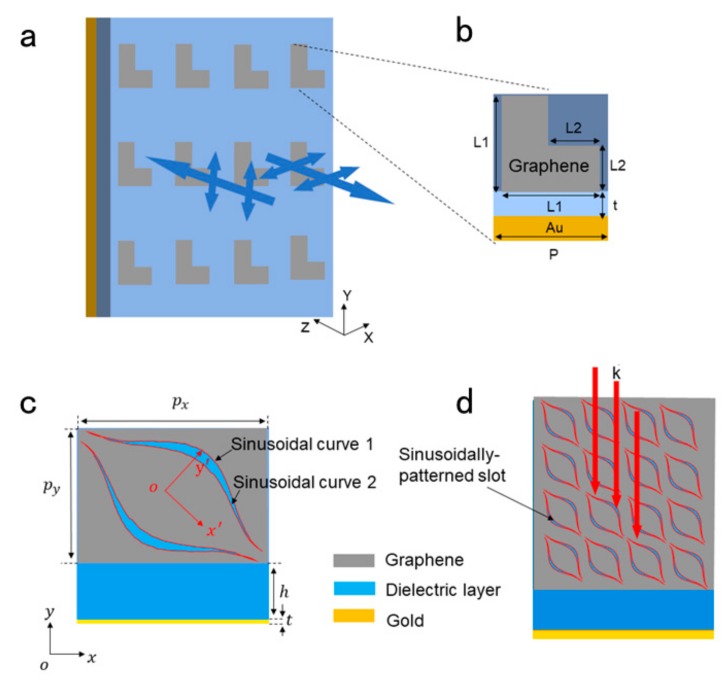
(**a**) Schematic of an ultrathin half-wave plate composed of an array of L-shaped graphene patches. (**b**) Square unit cell of the proposed design with the dimensions P = 3.6 μm, L_1_ = 2.4 μm, L_2_ = 1.2 μm, and t = 7.5 μm. (**c**) Unit cell of the proposed cross-polarization converter (CPC). Where the optimized parameters are, as follows:$\;p_{x}$ = 16 μm, $p_{y}$= 16 μm, h = 25 μm, and t = 0.4 μm. (**d**) Configurations of the proposed polarization converter.

**Figure 5 ijms-21-01608-f005:**
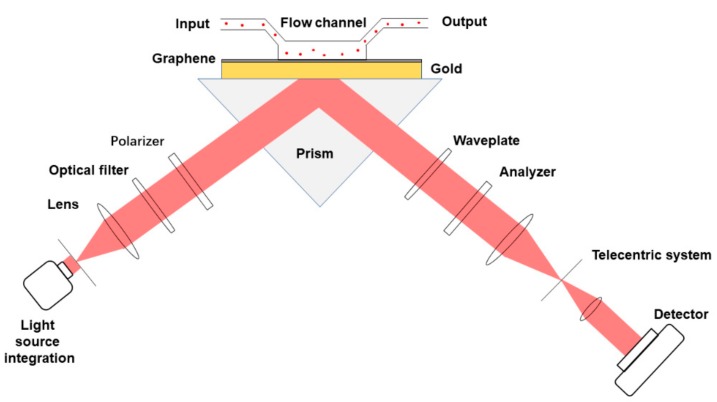
Schematic of the Surface Plasmon Resonance (SPR) polarization control method and layers.

**Figure 6 ijms-21-01608-f006:**
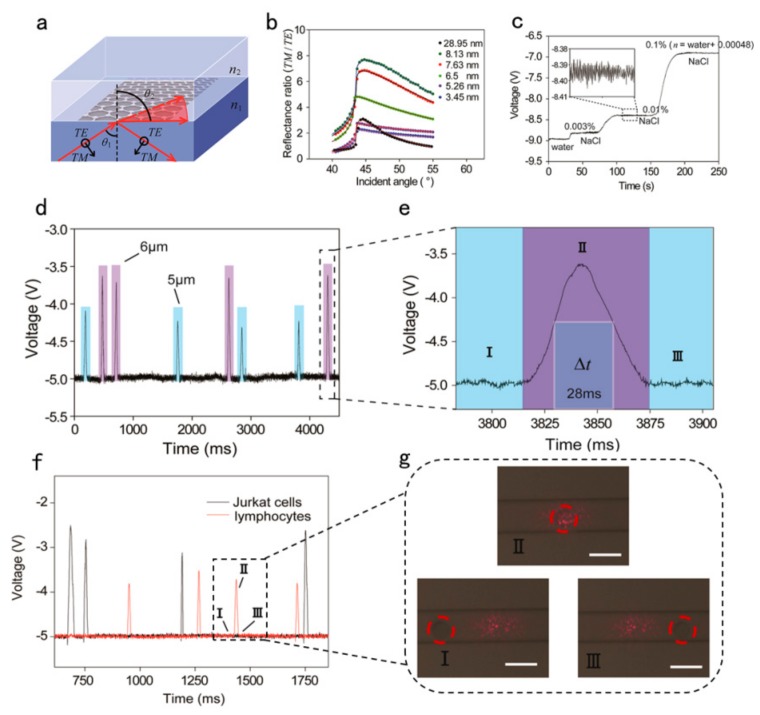
(**a**) Schematic of the enhanced sensitivity and resolution of refractive index (RI) sensing of high temperature reduced graphene oxide (h-rGO). (**b**) Angle-dependent reflectance ratio (TM/TE) plots of different thicknesses of h-rGO. (**c**) Real-time signal of different ultralow concentrations of NaCl solution. (**d**) Discrete time-dependent changes in voltage that correspond to PS microspheres as they roll across the h-rGO detection window. The light blue and light purple areas represent the discrete voltage signals of 5 and 6 μm PS microspheres, respectively. (**e**) Entire sensing process for a single PS microsphere detected by the graphene-based optical single-cell sensor (GSOCS) in which Δt represents the time changes for a PS microsphere that rolls across the h-rGO detection window. (**f**) Discrete time-dependent changes in voltage that correspond to a single lymphocyte or Jurkat cell as it rolls across the detection window. The black and red lines represent Jurkat cells and lymphocytes, respectively. (**g**) Microscopic images of the h-rGO detection window as lymphocytes roll across it. The scale bar is 15 μm, and the height of microfluidic channel is approximately 9 μm. Reproduced from [84] with permission of the American Chemical Society.

**Figure 7 ijms-21-01608-f007:**
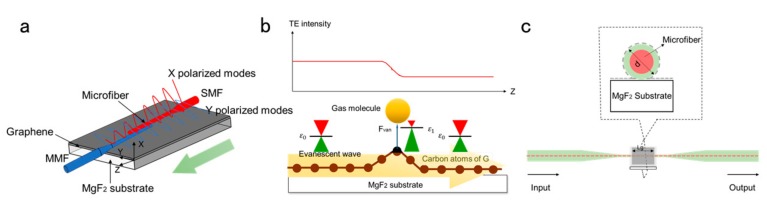
(**a**) Schematic of a hybrid graphene-waveguide coupled with a silica microfiber (GWMF) structure (the red curve shows TM mode and the blue curve shows the TE mode). (**b**) Working principle of GWMF for chemical vapor gas sensing. (**c**) Schematic diagram of the graphene multimode fiber interferometer (GMMI): Monolayer graphene film (honeycombs) coated on the microfiber set on the MgF_2_ substrate.

**Figure 8 ijms-21-01608-f008:**
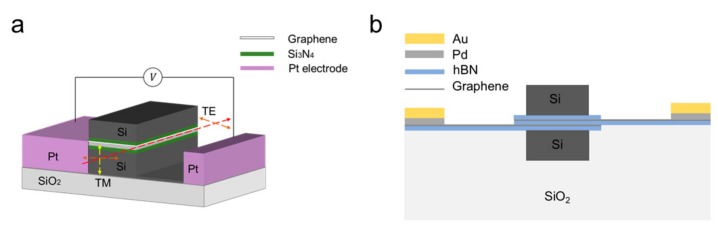
(**a**) Schematic of the polarization modulator. (**b**) Schematic of graphene-based optical phase modulator.

**Figure 9 ijms-21-01608-f009:**
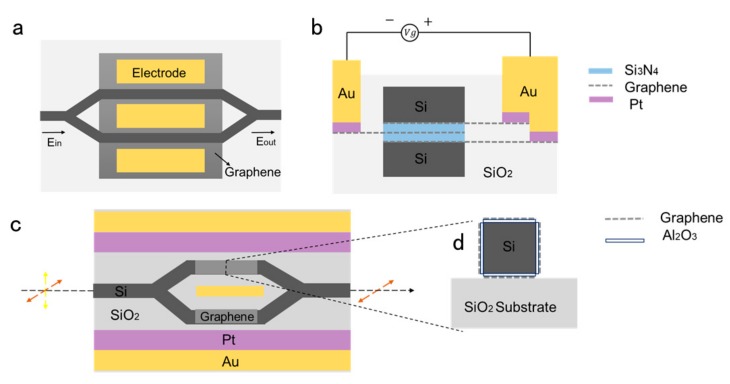
(**a**) Two-dimensional (2D) vertical view of the modulator and (**b**) 2D cross section of the modulator’s arm. One metal electrode is brought into contact with the first and third graphene layers. Another metal electrode is brought into contact with the second graphene layer. n_Si_=3.47, n_SiO2_=1.447, n_Si3N4_ =1.98. (**c**) The three-dimensional (3D) view of the proposed polarizer; (**d**) the schematic view of the proposed device.

**Figure 10 ijms-21-01608-f010:**
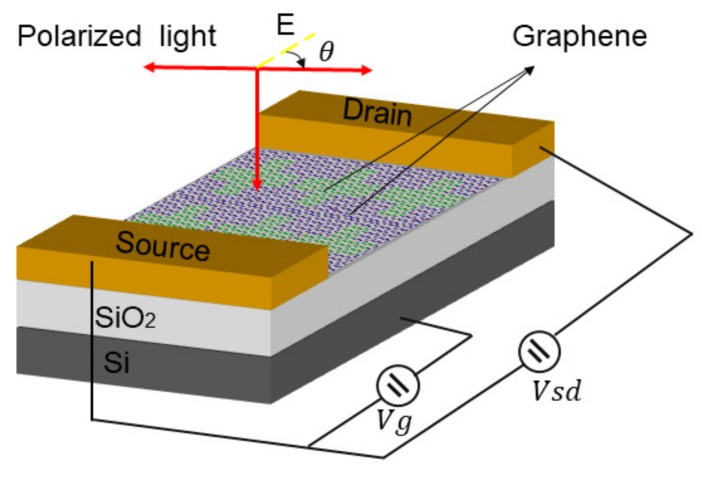
Schematic of graphene photodetector. The angle θ stands for polarization angle.

**Table 1 ijms-21-01608-t001:** Briefly describe the classification, working principle and advantages of graphene-based optical polarizing devices.

Optical Polarizing Devices	Work Principle	Advantage
**Graphene-based Polarizer**	Graphene in the intrinsic state can cause the transverse magnetic (TM) mode to be consumed through the leak mode and the TE mode to be retained. When the chemical potential of graphene is at the epsilon near zero (ENZ) point, the transverse electrical (TE) mode can generate plasmon resonance effect at the graphene-dielectric interface, thereby losing the TE mode and retaining the TM mode.	Strong compatibility, high extinction ratio, broadband, small volume and easy to integrate.
**Polarization Sensors Based on Graphene**	Under total internal reflection, graphene exhibits properties of enhanced polarization absorption and broadband absorption. The sensor uses the attenuated total reflection method to detect the refractive index change near the sensor surface.	High sensitivity, low sample consumption, fast measurement speed, no marking processing and real-time detection.
**Polarization Modulator Based on Graphene**	Applying a voltage changes the refractive index of graphene, which changes the absorption coefficient of graphene. Evanescent waves are generated when light travels through the waveguide. When graphene interacts with the evanescent wave, since the light absorption rate of graphene is controlled by the electric signal, the output light intensity is controlled by the electric signal, and finally the modulation effect is achieved.	High extinction ratio, high modulation efficiency, small loss and small volume.
**Polarization Photodetector Based on Graphene**	Graphene polarization-sensitive photodetector (GPSP) is composed of crossed graphene nanostructure with different dimensions deposited in SiO_2_ substrate. Localized surface plasmon is generated by patterned graphene structure with significantly enhanced incident light absorption. The tunability of photodetectors is achieved by changing gate voltage to modulate the chemical potential of graphene.	High light responsiveness, high specific detection rate, broadband and fast response speed.

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
