# Peer review of "Review of Polarization Optical Devices Based on Graphene Materials"

_ijms, 2020, doi:10.3390/ijms21051608_

Round 1

Reviewer 1 Report

Xing et al reviewed the recent development on polarization-based graphene devices. The timing and the soundness of the review are adequate. The authors did put a great effort into summarizing the findings. The manuscript may be accepted with minor changes. My comments are below,

  1. Many polarization devices described using single-layer CVD graphene for fabrication. Are there any studies investigating the role of the number of layers vs device performance. Please add them?
  2. The angle between graphene layers also controls the electronic properties of graphene (like magic-angle graphene). What would be the role of these interactions in terms of polarization?
  3. It would be nice if the authors can add a table summarizing all the efforts in the reviewed field, which may attract some readers. 

Author Response

Dear Sir,

Thank you very much for your comments.

The revisions are made according to your comments.

Point 1: Many polarization devices described using single-layer CVD graphene for fabrication. Are there any studies investigating the role of the number of layers vs device performance. Please add them? 

Response 1: Thanks for reviewer’s kind suggestion and we do agree with the reviewer. We consulted some related references on the study of graphene layers and polarization characteristics, and studied the polarization-dependent light absorption of graphene under total internal reflection. Added in section 3.2 lines 460-467 in the manuscript. It is found that as the number of graphene layers (one to four layers) increases, the reflectance of TE mode light gradually decreases, while the reflectance ratio (P/S) gradually increases. Therefore, using four-layer of graphene as the sensing material, the performance of the polarization sensor is better than that of the single-layer and double-layer of graphene. It is concluded that under total reflection, graphene absorbs more s-polarized light than p-polarized light. In addition, different polarizing devices correspond to different graphene layers. For example, Yang et al. designed a M-Z modulator based on a hybrid graphene silicon waveguide (GSW), which uses a three-layer graphene sheet. Please refer to section 4.2 for details.

Point 2: The angle between graphene layers also controls the electronic properties of graphene (like magic-angle graphene). What would be the role of these interactions in terms of polarization?

Response 2: Thank you for your reminding. At present, the graphene materials used in the polarizing devices mentioned in the review have not considered the fine corner control between graphene layers, and they are all randomly stacked thicknesses. The relationship between the fine corner control and polarization between graphene layers has not been clearly studied at present, and I believe that there will be further research in this area by researchers.

Point 3: It would be nice if the authors can add a table summarizing all the efforts in the reviewed field, which may attract some readers.

Response 3: Thanks to the reviewer for the kind reminding. At introduction line 90-93, we have compiled a table for a detailed description of all the work in the field of graphene-based polarizing devices.

Reviewer 2 Report

Comments and Suggestions for Authors In this paper, the research progress of polarizers, sensors, modulators and photodetectors based on the polarization characteristics of graphene is reviewed. The polarization dependent effect and broadband absorption of graphene and the enhancement effect of the interaction between graphene and light under the total reflection structure are emphasized and discussed. This paper provides a new direction and train of thought for the research of related devices based on the polarization characteristics of graphene. The overall proposed organization and section structure are consistent with the proposed contributions. It would be suitable for publication in the journal after applying revisions based on the following remarks. 1. At introduction line 25-27, the author mentioned that “Graphene, as a new type of carbon nanomaterial, is a single-layer benzene ring structure (hexagonal lattice honeycomb) two-dimensional crystal element composed of carbon atoms, and has only one carbon atom thickness”. However, it is better to add more details to describe the microstructure of graphene. For example, graphene is a monoatomic layer of SP2 carbon atom with two-dimensional hexagonal crystal structure (HCC). 2. In section 2 line 94-95, the author claimed that “Because of its adjustable electrical conductivity, graphene can obtain the properties of the medium 94 or metal under the influence of different applied voltage, which is described by the Kubo formula”. But no references are given. Citation of some related work would be helpful. 3. In section 2.1 line 111-114, it mentioned that “At the same time, due to its excellent electrical and optical properties, the chemical potential of graphene can be controlled by external voltage or chemical doping, which can be applied to fiber polarizer to effectively control the transmission of the TE mode and TM mode”. However, the discussion about the chemical potential of graphene controlled by chemical doping needs more discussion and literature support. 4. In section 3.2 line 453-455, the author mentioned that the region I-II-III in figure 6-f corresponds to phase I-II-III in figure 6-g. The size and flow of a single PS microsphere will affect the time variation 453 (Δt) of the PS microsphere rolling on the detection window. 454. It would be better to use a clearer picture in figure 6-g.

Author Response

Dear Sir,

Thank you very much for your comments.

The revisions are made according to your comments.

Point 1: At introduction line 25-27, the author mentioned that “Graphene, as a new type of carbon nanomaterial, is a single-layer benzene ring structure (hexagonal lattice honeycomb) two-dimensional crystal element composed of carbon atoms, and has only one carbon atom thickness”. However, it is better to add more details to describe the microstructure of graphene. For example, graphene is a monoatomic layer of SP2 carbon atom with two-dimensional hexagonal crystal structure (HCC).

Response 1: Thank you for your comments. At introduction line 25-27, we have made changes in the manuscript to replace "Graphene, as a new type of carbon nanomaterial, is a monoatomic layer of SP2 carbon atom with two-dimensional hexagonal crystal structure (HCC)" with "Graphene, as a new type of carbon nanomaterial, is a single-layer benzene ring structure (hexagonal lattice honeycomb) two-dimensional crystal element composed of carbon atoms, and has only one carbon atom thickness".

Point 2: In section 2 line 94-95, the author claimed that “Because of its adjustable electrical conductivity, graphene can obtain the properties of the medium 94 or metal under the influence of different applied voltage, which is described by the Kubo formula”. But no references are given. Citation of some related work would be helpful.

Response 2: Thank you for your kind reminding. Through Kubo formula, there has been in-depth research on the calculation of graphene conductivity. In section 2 line 99-104, we have provided some related work and cited some references in the manuscript for readers to study. For example, in 2008, Hanon used Kubo formula to calculate the surface conductivity of graphene, and concluded that when determining temperature, incident wavelength and scattering rate, the conductivity of graphene is only related to chemical potential. Newly cited references have been marked in red font, numbered 27 to 31.

Point 3: In section 2.1 line 111-114, it mentioned that “At the same time, due to its excellent electrical and optical properties, the chemical potential of graphene can be controlled by external voltage or chemical doping, which can be applied to fiber polarizer to effectively control the transmission of the TE mode and TM mode”. However, the discussion about the chemical potential of graphene controlled by chemical doping needs more discussion and literature support.

Response 3: We do agree with the reviewer's view. In section 2.1 line 122-132, we have provided supplementary knowledge of chemical doping to control the chemical potential of graphene in the manuscript. And a new reference is cited, number 36. For chemical doping, the n-doping state of graphene can be obtained by metal atom doping, and the p-doping state of graphene can be achieved by polymer molecules composed of N, O, F and other elements. It also provides precise methods for different and uniformly and reproducibly doped graphene p-type / n-type applications and controlling doping levels. Related reference is cited as support, serial number 36.

Point 4: In section 3.2 line 453-455, the author mentioned that the region I-II-III in figure 6-f corresponds to phase I-II-III in figure 6-g. The size and flow of a single PS microsphere will affect the time variation 453 (Δt) of the PS microsphere rolling on the detection window. 454. It would be better to use a clearer picture in figure 6-g.

Response 4: Thanks to the reviewer for pointing out this issue. In order to let the reader see the picture more clearly, we have enlarged the original picture. In addition, the cells are marked in the picture to make the reader more clearly see the flow of cells through the entire detection window.
